# Prevalence of Overweight and Obesity among People Aged 18 Years and Over between 2013 and 2018 in Hunan, China

**DOI:** 10.3390/ijerph17114048

**Published:** 2020-06-05

**Authors:** Junjie Hua, Lingling Zhang, Deyue Gao, Yun Huang, Peishan Ning, Peixia Cheng, Yingzi Li, Guoqing Hu

**Affiliations:** 1Department of Epidemiology and Health Statistics, Hunan Provincial Key Laboratory of Clinical Epidemiology, Xiangya School of Public Health, Central South University, Changsha 410078, China; huajunjie@csu.edu.cn (J.H.); gaodeyue@csu.edu.cn (D.G.); ningpeishan@csu.edu.cn (P.N.); chengpeixia@csu.edu.cn (P.C.); yingzi@csu.edu.cn (Y.L.); 2Department of Nursing, College of Nursing and Health Sciences, University of Massachusetts Boston, Boston, MA 021125, USA; Lingling.Zhang@umb.edu; 3Department of Occupational and Environmental Health, Xiangya School of Public Health, Central South University, Changsha 410078, China; 803110@csu.edu.cn

**Keywords:** overweight, obesity, prevalence, China

## Abstract

**Background:** Recent overweight and obesity prevalence data are lacking for China. **Methods:** Data were from provincially representative surveys conducted in 2013 and 2018 in Hunan Province, China. Overweight and obesity were defined according to the Chinese standard. Complex sampling weights were considered in statistical analyses. 95% confidence interval (95% CI) of rate was calculated. **Results:** The overall prevalence rate between 2013 and 2018 significantly increased from 20.81% (95% CI: 17.68–23.95) to 26.97% (95% CI: 23.48–30.45) for overweight and from 4.09% (95% CI: 3.21–4.96) to 7.13% (95% CI: 5.10–9.15) for obesity in Hunan Province of China, respectively. Urban residents and males had higher crude prevalence rates of overweight and obesity than rural residents and females in 2013 and in 2018. Notably, the peak age groups of overweight and obesity both became younger between 2013 and 2018. After controlling for Engel coefficient, level of education and frequency of physical exercise per week, the overweight prevalence significantly increased in urban males aged 65+ (adjusted OR: 1.52) and rural males aged 45–54 years (adjusted OR: 1.52) and 65+ (adjusted OR: 1.88) and the obesity prevalence rate rose significantly in many groups (urban males: 18–24 years, 25–34 years, 35–44 years and 55–64 years; urban females: 25–34 years, 35–44 years, 45–54 years; rural males: 18–24 years, 25–34 years; rural females: 35–44 years, 55–64 years), with adjusted OR varying from 1.56 to 5.52. **Conclusion:** The adult prevalence rates of overweight and obesity significantly increased between 2013 and 2018 in Hunan Province, China. The increasing prevalence rates and varying prevalence changes across groups warrantee further research and policy interventions.

## 1. Introduction

Excess body weight (overweight and obesity) is a risk factor of chronic noncommunicable diseases [1] and with at least 2.8 million people dying each year due to overweight or obesity [2]. In the Integrated Chronic Disease Prevention and Control Programme developed by the World Health Organization (WHO), the first objective is to strengthen prevention and control of chronic noncommunicable diseases by tackling the major risk factors [3]. China has made substantial efforts to address the increasing challenge of overweight and obesity in the past decade, including policies, guidelines and programs [4,5,6,7]. Naturally, it is important to monitor change in overweight and obesity prevalence rate to evaluate the effectiveness of these efforts.

Several publications reported national or local changes in overweight and obesity prevalence of children and adolescents or adults in China [8,9,10,11]. But none reported prevalence data after 2015 for all demographic groups (urban and rural areas, males and females, different age groups).

The National Health Service Household Interview Survey of China is a survey designed to comprehensively assess the status of resident health and utilization of health services. Since 1993, the national survey is conducted every five years. Hunan Province adopts the national survey protocol to conduct its own survey from 2013 and now has completed two round surveys in 2013 and 2018. Using the two-round survey data, this study examined change in overall and subgroup overweight and obesity prevalence rates between 2013 and 2018 in Hunan Province, China.

## 2. Materials and Methods

### 2.1. Data Sources

Hunan Province is located in the central area of China (north latitude: 24°38′–30°08′; east longitude: 108°47′–114°15′) (Figure 1) and has about 68.99 million resident population in 2018 [12]. Data were from the two-round health service household interview surveys of Hunan Province, China, which were conducted in 2013 and 2018 by the Center for Health Education and Information Statistics, Health Commission of Hunan Province. Two-round surveys adopted the same sampling framework. Study participants were selected at random using a multi-stage stratified cluster sampling. First, 14 municipalities of Hunan Province were divided into urban and rural sampling points at random (seven for urban points and seven for rural points); one district/county was further chosen from each sampling point randomly (district for urban sampling points and country for rural sampling points). Secondly, the researchers randomly chose 5 towns from each selected district/county. Thirdly, two communities/villages from each selected town (community for district towns and village for rural towns). Lastly, 60 households were selected randomly in each selected community/village and all family members of these 60 households were invited to participate in the survey. Oral consents were obtained before the face-to-face interview for all participants (including adult guardians of children under 10 years old). All data collectors received the standard training organized by the Health Commission of Hunan Province. Strict quality control measures were taken, including field inspection at each sampling district/county, choosing 5% of surveyed households to do a second survey through visiting or calling the households by independent data collectors, use of standard data entry software and implementation of standard logic checking. The survey items used in this study are listed in the Appendix A).

### 2.2. Outcome Measure

According to the national criteria of weight (Standard NO: WS/T 428-2013) issued by the National Health Commission of People’s Republic of China [13], we used a questionnaire to obtain the height and weight of the respondents to calculate the body mass index (BMI) and divided the respondents aged 18 years and over into four groups—underweight (<18.5), normal (18.5–23.9), overweight (24–27.9) and obesity (≥28.0).

### 2.3. Sociodemographic Variables

In terms of data availability, we considered three sociodemographic variables in our analysis—place (urban vs. rural), sex (male vs. female), age (18–24 years, 25–34 years, 35–44 years, 45–54 years, 55–64 years and ≥65 years).

### 2.4. Covariates

Based on previous reports [14,15] and data availability, we selected Engel coefficient, level of education and frequency of physical exercise as covariates of multivariate analyses.

Engel coefficient was calculated as the ratio of food expenditure by total expenditure. According to the recommendation by the Food and Agriculture Organization of the United Nations [16], we classified the surveyed households into five strata as follows—stratum one (>0.59), stratum two (0.50–0.59), stratum three (0.40–0.49), stratum four (0.30–0.39), stratum five (<0.30).

Level of education was combined into three groups—primary school and below, junior high school, high school and above.

Frequency of physical exercise per week was divided into 5 categories—6 times and more, 3–5 times, 1–2 times, less than once and never.

### 2.5. Statistical Analysis

Rao-Scott *χ*^2^ test was used to test prevalence rate changes between 2013 and 2018 and across place, sex and age group. Multivariate logistic regression models were fitted to examine place-, sex- and age-specific prevalence changes between 2013 and 2018 after controlling for Engel coefficient, level of education and frequency of physical exercise per week. Sampling weights were considered in all statistical analyses. SAS 9.4 (SAS Institute Inc., Cary, NC, USA) and GraphPad Prism 8.0 (GraphPad Software, San Diego, CA, USA) were used to complete statistical analyses. *p* < 0.05 was considered statistically significant. All *p* values were based on two-tailed statistical tests. We followed the Strengthening the Reporting of Observational Studies in Epidemiology (STROBE) Statement to report this study.

### 2.6. Ethical Statement

All respondents gave their informed consent for inclusion before they participated in the study. The study was conducted in accordance with the Declaration of Helsinki and the survey protocol was approved by the Hunan Health Commission. The study analysis was approved by the Medical Ethics Committee of Central South University on 24 February 2020 (No. XYGW-2020-46).

## 3. Results

### 3.1. Characteristics of Respondents in the Two Interviews

There were 24,282 valid responses out of 24,286 in 2013 and 22,530 out of 22,532 in 2018. 80% of respondents were 18 years and older in both surveys. The percentage of urban residents increased from 23.46% to 30.62% between 2013 and 2018 although the change was not statistically significant (Table 1). The ratio of males/females was much closer to 1.0 and remained nearly unchanged between 2013 and 2018. Compared in 2013, the age structure became a little older in 2018. The percentages of residents with overweight and obesity increased substantially between 2013 and 2018.

### 3.2. Crude Prevalence Rates

The crude prevalence of overweight increased from 20.81% (95% CI: 17.68–23.95) in 2013 to 26.97% (95% CI: 23.48–30.45) in 2018 (Table 1, Figure 2). Subgroup analyses by place and sex showed significant increases in overweight prevalence rate in all subgroups between 2013 and 2018 (urban area: from 24.73% to 29.14%; rural area: from 19.61% to 26.01%; males: from 22.00% to 28.47%; females: from 19.67% to 25.56%).

The overall prevalence of obesity rose from 4.09% (95% CI: 3.21–4.96) in 2013 to 7.13% (95% CI: 5.10–9.15) in 2018 (Table 1, Figure 2). Between 2013 and 2018, the crude prevalence rate of obesity increased in both urban area and rural area (from 4.68% to 7.52% and from 3.90% to 6.95%) and both males and females (from 4.23% to 7.71% and from 3.94% to 6.58%).

### 3.3. Place-, Sex- and Age-Specific Prevalence Rates

Table 2 presents place-, sex- and age-specific overweight prevalence rates in 2013 and in 2018. The prevalence rate differences across place and sex did not follow a consistent pattern for all age groups in both years. Notably, the age-specific overweight prevalence peaked at the middle age groups (35–44 years, 45–54 years or 55–64 years) for both urban area and rural area and for both males and females. After adjusting for Engel coefficient, level of education and frequency of physical exercise per week, there were three subgroups demonstrating significant prevalence increases between 2013 and 2018—urban males aged 65 and above (adjusted OR = 1.52, 95% CI: 1.11–2.08), rural males aged 45–54 years and aged 65 years and above (adjusted OR = 1.52, 95% CI: 1.05–2.21; adjusted OR = 1.88, 95% CI: 1.09–3.23).

The obesity prevalence differences across place, sex and age group were complex (Table 3). Notably, the peak age group of obesity prevalence became younger between 2013 and 2018; the peak age group changed to be 25–34 years for both urban and rural males and rural females and became 45–54 years for urban females in 2018. Many age groups experienced substantial and significant increases in obesity prevalence rate after adjusting for Engel coefficient, level of education and frequency of physical exercise per week, including—urban males—18–24 years (adjusted OR = 3.56), 25–34 years (adjusted OR = 2.40), 35–44 years (adjusted OR = 1.64) and aged 55–64 years (adjusted OR = 1.82);

urban females—25–34 years (adjusted OR = 3.00), 35–44 years (adjusted OR = 2.06) and 45–54 years (adjusted OR = 1.62);

rural males—18–24 years (adjusted OR = 3.96) and 25–34 years (adjusted OR = 5.52);

rural females—35–44 years (adjusted OR = 1.56) and 55–64 years (adjusted OR = 1.91).

## 4. Discussion

### 4.1. Primary Findings

Compared to previous publications related to China [8,9,10,11], this study updated the latest overweight and obesity prevalence rates of residents aged 18 years and older and generated four important findings. Firstly, the crude prevalence rates increased substantially between 2013 and 2018 for both overweight and obesity among people aged 18 years and over in Hunan Province, China. Secondly, the crude prevalence rates of overweight and obesity were generally higher in urban area and in men than in rural area and in women. Thirdly, overweight and obesity tended to occur in younger populations between 2013 and 2018. Last, many place-, sex-and age-specific prevalence rates increased significantly during the study time period.

### 4.2. Interpretation of Findings

To our knowledge, this is the first published study using post-2015 data to report increases in overweight and obesity prevalence rates in Chinese adults. The results were similar with the reports from the United States for 2013 and 2018 (age-adjusted obesity prevalence of increasing from 37.7% to 42.4%) [17,18]. The recent increasing prevalence rates primarily reflect the combined effects of unhealthy diet, lack of physical activity and long sedentary time. Over the past five decades, food consumption has risen rapidly in China, with daily calorie intake per capita increasing from 1421 kcal in 1961 to 2950 kcal in 2017; daily diet has undergone significant changes. From 1961 to 2017, Chinese diet pattern has changed dramatically. In 1961, starchy foods such as cereals and tubers were the main components of the Chinese population’s diet, accounting for 78% of energy intake; white meat was least consumed, only 13 kcal per day. Since 1983, the intake of animal foods such as meat and eggs has increased steadily. By 2017, the average calorie intake of red meat, fruits & vegetables increased 16-fold and 9-fold, respectively. In contrast, the daily calorie intake of cereals, roots and tubers dropped by half [19]. This shift of Chinese diet pattern was associated with an increased risk of overweight and obesity [20,21]. The proportion of Chinese adults reaching the recommended criteria for leisure-time physical activity is low, although it rose from 17.2% to 22.8% between 2000 and 2014 [22]. The proportion of Chinese adults spending more than 4 h sedentarily each day increased from 35.4% in 2002 to 43.0% in 2012 [23].

Higher prevalence rates in urban area and in males are probably related to sharp decreases in physical activities, changes in daily diet and increased psychological stress accompanied with fast-raced life [24,25,26]. Urban residents have more sedentary time and spend less time in physical activities than rural residents; the growing popularity of television and computer games also decreases active leisure, especially among children and young adults [26,27]. High-density population, convenient public transportation and scarcity of parks in urban areas make residents do less physical exercise than in rural areas [28]. Urbanization also contributes to changing consumer preferences, purchase habits, food environment and a shift towards Western-style diet [25,26]. Supermarkets and processed foods are becoming increasingly popular in the retail revolution in developing countries and may lead to overweight and obesity, especially among urban consumers [25]. Compared to females, males prefer eating high-calorie foods (fish and meat, etc.) [29,30,31] and are more likely to buy foods from convenience stores [32]. In addition, many females are particularly concerned about with their body image; they may be more likely to adopt a healthy diet than males [33,34].

The move of peak age group to younger adults between 2013 and 2018 are striking. A previous systematic review showed that people of all ages in successive cohorts seem to be gaining weight, with more gains in adults age 20–40 years [35]. So far, there are no solid evidence to explain the phenomenon. It is likely related to the rapid development of E-commerce in China. China’s E-commerce transactions have experienced an explosive growth from 2012 to 2016, with an annual rate of 38.2% [36]. E-commerce transactions not only significantly increase the likelihood of users to access unhealthy foods but also substantially reduce users’ time on physical activities. Notably, young adults ages 25–34 years were the most common users of E-commerce transactions, account for 33% of all users in 2019 [37].

### 4.3. Policy Implications

Our findings have two important implications. First, these results highlight the increasing threat of overweight and obesity in China. Although the results were based on a provincial sample, they may reflect the realities of many other provinces. Because the socio-economic development follows a similar mode and yields a common impact on residents’ life behaviors for all provinces in China (e.g., eating, physical activities). No doubt, more and stronger interventions should be taken by the government to curb the rising trend in the future. For example, it is worthwhile to promote the diet pattern according to the traditional principles and key elements of Mediterranean diet [38,39]. Second, heavier overweight and obesity burden in urban residents and in males, as well as the move of heaviest burden to younger adults, warrant further research to interpret these phenomena and develop intervention programs to reduce disparities across place (urban/rural) and sex and to prevent the growing prevalence rates in younger adults.

### 4.4. Study Limitations

This study has three major limitations. First, due to the unavailability of data, we merely used data from two repeated cross-sectional surveys in 2013 and 2018 to approximately measure changes over the five years, missing possible changes in the study time period. Second, we did not explore the reasons for the move of peak age group to younger adults and interpret urban-rural and male-female differences in overweight and obesity prevalence rates because of lack of relevant data. Last, our findings were based on data from the sample of Hunan Province and may not be representative for the whole country and other provinces in terms of potential variations across provinces in mainland of China [40].

## 5. Conclusions

The prevalence rates increased substantially between 2013 and 2018 for both overweight and obesity among adults in Hunan Province, China. Urban residents and men were at comparatively higher risks of overweight and obesity than rural residents and women in both 2013 and 2018. The peak age of overweight and obesity tended to be younger between 2013 and 2018. Further research is needed to investigate the possible reasons for the move to younger age groups. Evidence-based interventions should be implemented to curb the increasing overweight and obesity prevalence rates among residents in Hunan Province, China.

## Figures and Tables

**Figure 1 ijerph-17-04048-f001:**
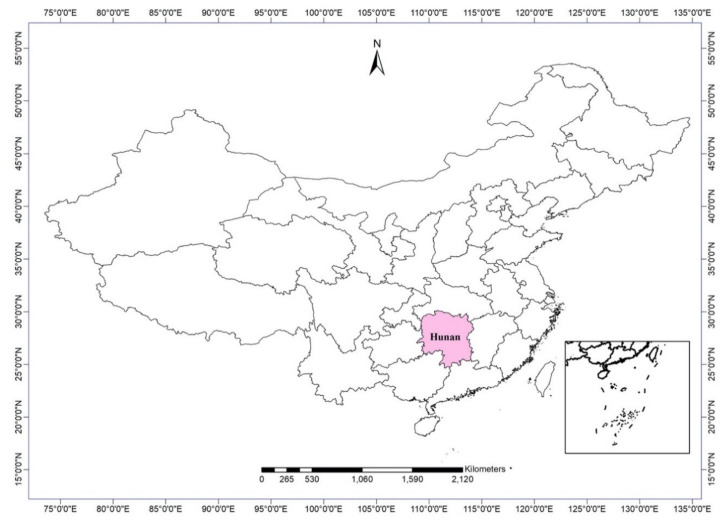
Geographical position of Hunan Province, China.

**Figure 2 ijerph-17-04048-f002:**
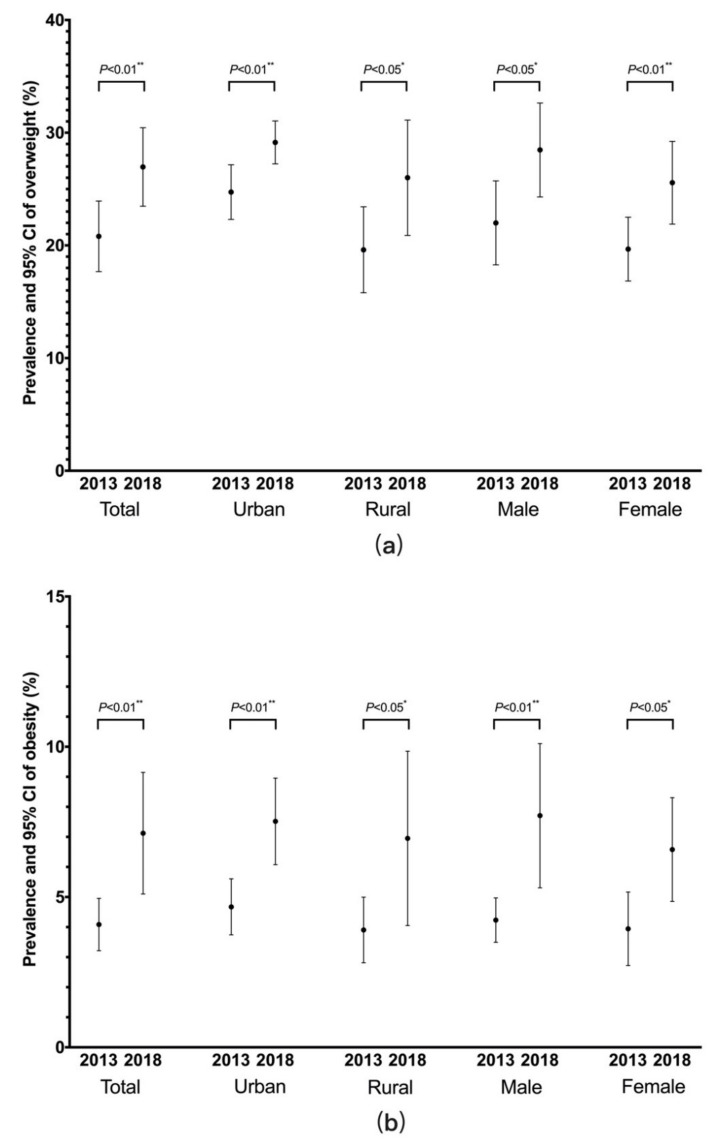
(**a**) Prevalence rates of overweight among inhabitants aged 18 years and older between 2013 and 2018 by place and sex in Hunan Province, China; (**b**) Prevalence rates of obesity among inhabitants aged 18 years and older between 2013 and 2018 by place and sex in Hunan Province, China. 95% CI: 95% confidence interval.

**Table 1 ijerph-17-04048-t001:** Characteristics of respondents in health service household interviews of 2013 and 2018 in Hunan Province, China.

Variable	2013		2018		*p*
	Number	Percentage (95% CI)	Number	Percentage (95% CI)	
Total	19,387	100.00	17,959	100.00	
Place ^#^					0.70
Urban	9929	23.46 (4.52–42.40)	9214	30.62 (6.32–54.92)	
Rural	9458	76.54 (57.60–95.48)	8745	69.38 (45.08–93.68)	
Sex					0.15
Male	9504	49.00 (48.29–49.71)	8749	48.20 (47.29–49.12)	
Female	9883	51.00 (50.29–51.71)	9210	51.80 (50.88–52.71)	
Age group (years)					<0.01
18–24	1164	6.58 (5.44–7.72)	654	2.79 (2.41–3.17)	
25–34	2463	12.59 (10.28–14.90)	2179	10.81 (9.31–12.32)	
35–44	3297	17.07 (15.62–18.52)	2370	12.57 (11.35–13.78)	
45–54	4377	22.51 (20.12–24.91)	4552	26.18 (24.37–28.00)	
55–64	4267	21.98 (20.19–23.76)	3865	22.52 (20.77–24.27)	
≥65	3819	19.27 (16.23–22.30)	4339	25.13 (23.05–27.22)	
BMI group					<0.01
Underweight	2222	11.75 (9.61–13.89)	1811	10.26 (7.57–12.95)	
Normal	12,234	63.35 (60.49–66.22)	10,392	55.65 (52.66–58.65)	
Overweight	4113	20.81 (17.68–23.95)	4609	26.97 (23.48–30.45)	
Obesity	818	4.09 (3.21–4.96)	1147	7.13 (5.10–9.15)	

95% CI: 95% confidence interval. BMI: Body mass index. ^#^: The sampling framework treated population living in sample counties as rural residents.

**Table 2 ijerph-17-04048-t002:** Place-, sex- and age-specific overweight prevalence rates among inhabitants between 2013 and 2018 in Hunan Province, China.

Place	Sex	Age Group (Years)	Prevalence (95% CI)		Unadjusted OR	Adjusted OR ^a^
			2013	2018	(95% CI)	(95% CI)
Urban	Male	18–24	18.39 (16.40–20.37)	13.23 (7.23–19.24)	0.68 (0.40–1.13)	0.69 (0.35–1.35)
		25–34	26.84 (18.85–34.82)	30.96 (25.62–36.30)	1.22 (0.78–1.93)	1.13 (0.69–1.86)
		35–44	33.81 (30.26–37.37)	39.29 (33.58–45.00)	1.27 (0.96–1.67)	1.18 (0.98–1.43)
		45–54	36.71 (27.73–45.68)	37.12 (34.22–40.03)	1.02 (0.69–1.50)	0.98 (0.68–1.42)
		55–64	28.72 (24.52–32.91)	35.09 (27.10–43.08)	1.34 (0.91–1.98)	1.29 (0.84–1.97)
		≥65	24.76 (18.98–30.54)	34.37 (31.30–37.44)	1.59 (1.15–2.20) **	1.52 (1.11–2.08) **
	Female	18–24	5.38 (1.59–9.18)	10.93 (6.03–15.82)	2.16 (0.91–5.08)	1.57 (0.58–4.25)
		25–34	14.99 (12.63–17.34)	12.96 (9.12–16.79)	0.84 (0.58–1.22)	0.79 (0.54–1.16)
		35–44	19.35 (14.47–24.22)	22.86 (20.03–25.68)	1.24 (0.88–1.73)	1.31 (0.94–1.81)
		45–54	24.63 (21.21–28.05)	27.10 (22.44–31.76)	1.14 (0.86–1.51)	1.08 (0.75–1.56)
		55–64	26.36 (24.62–28.09)	29.70 (22.54–36.86)	1.18 (0.84–1.65)	1.16 (0.82–1.63)
		≥65	21.68 (16.30–27.06)	26.82 (19.82–33.83)	1.32 (0.84–2.09)	1.20 (0.81–1.78)
Rural	Male	18–24	18.62 (9.61–27.63)	12.80 (4.48–21.13)	0.64 (0.26–1.59)	0.76 (0.31–1.87)
		25–34	21.09 (15.54–26.65)	21.21 (18.24–24.18)	1.01 (0.70–1.44)	1.03 (0.63–1.68)
		35–44	26.94 (23.08–30.79)	28.60 (25.06–32.15)	1.09 (0.85–1.39)	1.02 (0.79–1.32)
		45–54	25.49 (19.14–31.84)	34.36 (25.40–43.32)	1.53 (0.93–2.51)	1.52 (1.05–2.21) *
		55–64	16.82 (10.72–22.92)	24.38 (20.67–28.10)	1.59 (1.01–2.52) *	1.52 (0.91–2.54)
		≥65	10.35 (7.56–13.15)	19.67 (11.13–28.21)	2.12 (1.18–3.83) *	1.88 (1.09–3.23) *
	Female	18–24	12.26 (9.12–15.39)	6.67 (0.00–14.47)	0.51 (0.15–1.75)	0.91 (0.22–3.76)
		25–34	11.30 (8.69–13.91)	15.87 (11.65–20.10)	1.48 (1.00–2.19) *	1.14 (0.62–2.10)
		35–44	22.26 (18.47–26.04)	33.49 (22.77–44.22)	1.76 (1.06–2.91) *	1.83 (0.78–4.27)
		45–54	26.06 (20.30–31.82)	33.65 (27.23–40.07)	1.44 (0.97–2.14)	1.38 (0.90–2.12)
		55–64	19.81 (14.62–25.01)	27.11 (22.13–32.09)	1.51 (1.02–2.23) *	1.36 (0.93–1.97)
		≥65	15.48 (11.06–19.89)	19.79 (17.29–22.29)	1.35 (0.94–1.92)	1.20 (0.81–1.77)

OR: odds ratio; 95% CI: 95% confidence interval. ^a^: Adjusted odds ratios were estimated after adjusting for Engel coefficient, level of education and frequency of physical exercise per week. * *p* < 0.05; ** *p* < 0.01.

**Table 3 ijerph-17-04048-t003:** Place-, sex- and age-specific obesity prevalence rates among inhabitants between 2013 and 2018 in Hunan Province, China.

Place	Sex	Age Group (Years)	Prevalence (95% CI)		Unadjusted OR	Adjusted OR ^a^
			2013	2018	(95% CI)	(95% CI)
Urban	Male	18–24	3.63 (1.57–5.69)	7.24 (0.00–15.09)	2.07 (0.60–7.23)	3.56 (1.21–10.43) *
		25–34	6.74 (5.42–8.05)	14.79 (10.58–19.01)	2.40 (1.65–3.50) **	2.40 (1.64–3.51) **
		35–44	6.96 (5.52–8.40)	11.73 (9.60–13.87)	1.78 (1.33–2.37) **	1.64 (1.19–2.25) **
		45–54	6.49 (3.88–9.10)	8.72 (6.10–11.33)	1.38 (0.82–2.31)	1.52 (0.87–2.64)
		55–64	4.17 (3.09–5.24)	7.54 (4.28–10.80)	1.88 (1.12–3.14) *	1.82 (1.01–3.28) *
		≥65	4.70 (2.76–6.65)	4.47 (2.14–6.80)	0.95 (0.49–1.84)	1.05 (0.53–2.07)
	Female	18–24	2.58 (0.00–5.67)	1.26 (0.00–3.60)	0.48 (0.06–4.14)	0.21 (0.03–1.60)
		25–34	1.51 (0.89–2.14)	5.46 (3.55–7.36)	3.76 (2.21–6.39) **	3.00 (1.92–4.71) **
		35–44	2.01 (1.08–2.94)	4.00 (1.90–6.11)	2.03 (1.02–4.05) *	2.06 (1.02–4.19) *
		45–54	4.86 (2.85–6.87)	8.47 (6.24–10.71)	1.81 (1.10–2.98) *	1.62 (1.03–2.54) *
		55–64	4.86 (2.66–7.06)	5.64 (3.51–7.77)	1.17 (0.65–2.11)	1.22 (0.65–2.30)
		≥65	6.15 (4.38–7.91)	7.94 (5.94–9.94)	1.32 (0.89–1.95)	1.21 (0.85–1.72)
Rural	Male	18–24	1.74 (0.00–3.79)	6.97 (1.57–12.38)	4.23 (1.05–16.94) *	3.96 (1.00–15.65) *
		25–34	3.49 (1.72–5.26)	12.38 (7.39–17.37)	3.90 (2.01–7.60) **	5.52 (2.43–12.52) **
		35–44	6.10 (3.05–9.14)	9.40 (6.45–12.36)	1.60 (0.87–2.93)	1.39 (0.63–3.06)
		45–54	6.06 (4.17–7.95)	9.16 (5.03–13.30)	1.56 (0.88–2.76)	1.54 (0.93–2.57)
		55–64	2.64 (1.05–4.23)	8.11 (1.99–14.23)	3.25 (1.22–8.67) *	2.27 (0.79–6.50)
		≥65	1.55 (0.84–2.26)	1.79 (1.22–2.36)	1.16 (0.68–1.99)	1.41 (0.77–2.59)
	Female	18–24	4.46 (0.16–8.77)	3.46 (0.00–7.82)	0.77 (0.16–3.70)	2.88 (0.61–13.64)
		25–34	4.22 (0.00–8.64)	9.58 (4.79–14.37)	2.41 (0.75–7.75)	2.48 (0.97–6.38)
		35–44	4.16 (2.85–5.47)	5.70 (3.62–7.78)	1.39 (0.86–2.26)	1.56 (1.00–2.45) *
		45–54	4.85 (3.95–5.75)	7.57 (3.57–11.57)	1.61 (0.90–2.86)	1.67 (0.81–3.48)
		55–64	3.17 (1.90–4.44)	7.14 (3.90–10.39)	2.35 (1.27–4.34) **	1.91 (1.18–3.09) **
		≥65	3.42 (0.93–5.91)	4.96 (3.41–6.52)	1.47 (0.67–3.23)	1.19 (0.59–2.41)

OR: odds ratio; 95% CI: 95% confidence interval. ^a^: Adjusted odds ratios were estimated after adjusting for Engel coefficient, level of education and frequency of physical exercise per week. * *p* < 0.05; ** *p* < 0.01.

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
