# Peer review of "Prevalence of Overweight and Obesity among People Aged 18 Years and Over between 2013 and 2018 in Hunan, China"

_ijerph, 2020, doi:10.3390/ijerph17114048_

Round 1

Reviewer 1 Report

The article  does not seem to be particularly original but it is related to the journal's topics and it contributes to our knowledge of overweight and obesity in Hunan citizens, China. In fact, evidence show that in the last decade, the spread of obesity has been increasing at an alarming rate, especially in China.

The text is exhaustive.

Some sections should be reconsidered and improved.

The "Materials and methods" section requires figures indicating the geographical position of the studied area of ​​China.

Was a questionnaire used for the interviews? Has it been validated? Please, indicate the criteria and choice of the different items and attach the questions used.

In the  introduction or in the discussion, the authors do not specify the type of diet followed in Eastern countries compared to Western and Mediterranean countries (Chinese food, western diet, Mediterranean diet). Which foods are mainly consumed in China? A comment on this aspect could be interesting.

To this end, I recommend reading and / or citing the following works:

Anna Herforth, Mary Arimond, Cristina Álvarez-Sánchez, Jennifer Coates, Karin Christianson, Ellen MuehlhoffA - Global Review of Food-Based Dietary Guidelines  -  Advances in Nutrition, Volume 10, Issue 4, July 2019, Pages 590–605, https://doi.org/10.1093/advances/nmy130

F Bagordo, T Grassi, F Serio, A Idolo, A De Donno -Dietary habits and health among university students living at or away from home in southern Italy - Journal of Food & Nutrition Research, 2013

S. D’Innocenzo, C Biagi,  M.Lanari  -Obesity and the Mediterranean Diet: A Review of Evidence of the Role and Sustainability of the Mediterranean Diet -  Nutrients. 2019 Jun; 11(6): 1306. Published online 2019 Jun 9. doi: 10.3390/nu11061306

Reviewer 2 Report

This paper presents an analysis of the influence of various sociodemographic factors on the change in the occurrence of excessive body weight in adults from the Chinese province of Hunan and for this reason it is important for epidemiological studies in this region. It is worth extending this analysis to assess the impact of nutritional factors on the increase in overweight and obesity in people living in this region.

The description of the methods lacked information about whether the results of body mass and height were obtained on the basis of measurements or surveys.

Reviewer 3 Report

IJERPH-813749 Prevalence of overweight and obesity among person aged 18 years and over between 2013 and 2018 in Hunan, China.

This submission reports the results of summary of two provincial studies, five years apart to examine overweight and obesity prevalence and trends among Chinese adults.  The authors used appropriately-weighted sampled data to minimize sampling bias. A review of the English is needed. Under study limitations, it appears that the authors desire to report that they did not attempt to complete an in-depth assessment of potential reasons for the obesity peaks among the youngest age groups but likely due to working to phrase the statement in English this limitation is mis-stated. Tables and figures are appropriate. Overall, this is a clearly written submission detailing recent figures for prevalence and rise in overweight and obesity among adults in China. A review of the English is needed, and the submission will be an interesting contribution to the literature.
